# CAT: A Metric-Driven Framework for Analyzing the Consistency-Accuracy Relation of LLMs under Controlled Input Variations

## Abstract

We introduce CAT, a framework designed to evaluate and visualize the *interplay* of *accuracy* and *response consistency* of Large Language Models (LLMs) under controllable input variations, using multiple-choice (MC) benchmarks as a case study. Current evaluation practices primarily focus on model capabilities such as accuracy or benchmark scores and, more recently, measuring consistency is being considered an essential property for deploying LLMs in high-stake, real-world applications. We argue in this paper that although both dimensions should still be evaluated independently, their inter-dependency also need to be considered for a more nuanced evaluation of LLMs. At the core of CAT are the *Consistency-Accuracy Relation (CAR)* curves, which visualize how model accuracy varies with increasing consistency requirements, as defined by the *Minimum-Consistency Accuracy (MCA)* metric. We further propose the *Consistency-Oriented Robustness Estimate (CORE)* index, a global metric that combines the area and shape of the CAR curve to quantify the trade-off between accuracy and consistency. We present a practical demonstration of our framework across a diverse set of generalist and domain-specific LLMs, evaluated on multiple MC benchmarks. We also outline how CAT can be extended beyond MC tasks to support long-form, open-ended evaluations through adaptable scoring functions.

## 1 Introduction

Large Language Models (LLMs) have advanced rapidly in recent years, yet systematic evaluation practices seem to have kept pace. Most evaluation practices have focused on functional capabilities such as benchmark scores or task-specific accuracy while overlooking important non-functional properties (Aaron Grattafiori et al, 2024; DeepSeek-AI & Aixin Liu et al, 2024; Petrov et al., 2025). One such property is *response consistency*, considered here as the ability of the LLM to answer similar questions (or *prompts*, in current LLM-related jargon), which have the same correct answer, with answers similar to the correct answer. This is clearly an essential requirement for deploying LLMs in real-world, high-stake domains like healthcare, law, and finance, where input variations should not lead to inconsistent or contradictory outputs.

It is reasonable to expect that, depending on the task and the context, the prevalence of accuracy over consistency, or vice-versa, changes. However, there is a lack of tools and methods to allow developers and system deployment decision-makers to understand the nature of this trade-off for a given system. The framework described in this paper was created to address this methodological gap.

We thus present CAT (Consistency-Accuracy Toolkit), a comprehensive framework for evaluating the interaction between response consistency of LLMs and accuracy under input perturbations, with an initial focus on multiple-choice (MC) benchmarks. The core of the framework is the *Consistency-Accuracy Relation (CAR)* curve, which captures how model accuracy varies as we progressively increase the threshold for acceptable response consistency. This is formalized through the *Minimum-Consistency Accuracy (MCA)* metric, which defines accuracy conditioned on a minimum level of required consistency across input variants. To summarize the overall trade-off between consistency and accuracy, we introduce the *Consistency-Oriented Robustness Estimate (CORE)* index, a global metric that combines the area under the CAR curve with its deviation from an idealized perfectly

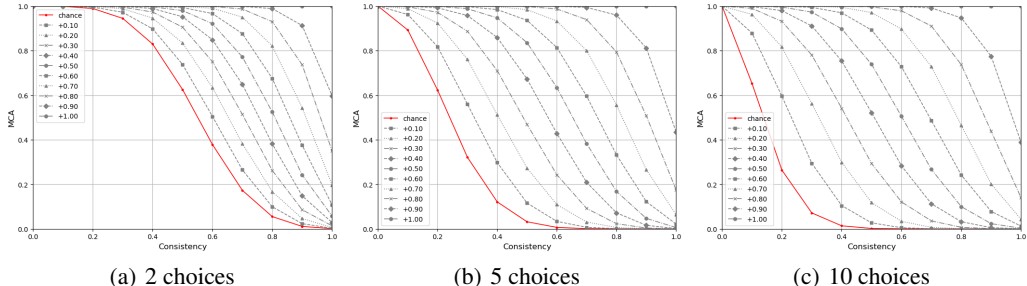

|     |     |     |
| :-: | :-: | :-: |
| (a) 2 choices | (b) 5 choices | (c) 10 choices |

Figure 1: CAR curves for synthetic models starting from a chance model to models with increasing bias toward the correct answer.

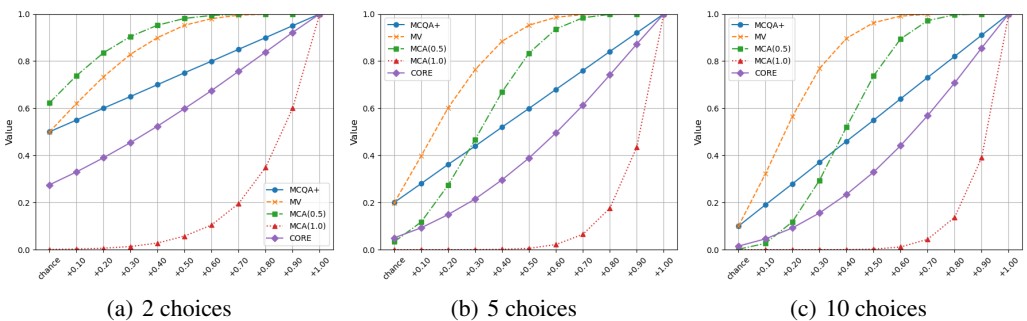

|     |     |     |
| :-: | :-: | :-: |
| (a) 2 choices | (b) 5 choices | (c) 10 choices |

Figure 2: Metric value growth across synthetic models. CORE emphasizes early gains and reduces bias towards chance.

consistent model. Together, our metrics provide a testbed to visualize and measure the interplay between accuracy and consistency, which is key for generative AI applications in real-world.

To facilitate the understanding of our approach, we consider in this paper only multiple-choice (MC) benchmarks which have emerged as a valuable and controllable way for studying the accuracy and consistency of LLMs. Their structured format allows researchers to systematically introduce meaning-equivalent input variations, such as simple changes in prompt phrasing, choice order, and formatting, and observe how these perturbations affect model behavior. Leveraging this setup, recent studies have revealed that LLMs are often highly sensitive to even minor changes in input structure (Habba et al., 2025; Wang et al., 2025; Nalbandyan et al., 2025). Notice that we do not explore in this paper issues related to *inference consistency*, where the ability to answer, under repeated trials, the exact same question or prompt with meaning-equivalent answers, such as examined in (Pinhanez et al., 2025), but our framework can easily handle such cases.

To provide an initial understanding of the framework proposed here, let us consider how three synthetic MC random-answering models of 2, 5, and 10 choices would be represented as CAR (Consistency-Accuracy Relation) curves and their MCA metric values. Figure 1 presents the CAR curves for the three synthetic random-answering experiments. We simulated models with varying levels of positive bias toward the correct answer, i.e. progressive levels of accuracy, and included a baseline chance model (marked in red) as a starting point, where all alternatives have the same probability of being sampled. We then introduced incremental biases of 0.1 in the probability of the correct alternative until it reaches 1.0. Each experiment varied in the number of choices, i.e. 2, 5, and 10, to highlight how the consistency-accuracy relationship evolves with task complexity. These curves reveal a non-linear relationship between consistency and accuracy, which becomes more pronounced as the number of choices increases.

Similarly, figure 2 shows how our proposed CORE index captures this non-linearity effectively, unlike two often used accuracy metrics: *MCQA+* (Pezeshkpour & Hruschka, 2024) which behaves almost linearly as model accuracy increases; and *MV* (Wang et al., 2024) that tends to saturate quickly, in a

behavior which is close to MCA(0.5), that computes accuracy with a very permissive consistency level. Notice that CORE offers a balance between the permissiveness of MCA(0.5) and the strictness MCA(1.0), providing realistic but not overly-punishing scores for the consistency-accuracy interplay. Additionally, CORE reduces bias towards chance-level performance, a limitation often overlooked in existing benchmark metrics. We define these metrics in detail in Section 3.

We present a practical demonstration of CAT on eight LLMs across four MC benchmarks, including both generalist and domain-specific models. Our results show that our approaches provide a complementary and more nuanced view of model behavior, enabling better-informed model selection. We also discuss how the framework can be extended to long-form and open-ended tasks through adaptable scoring functions.

The main contributions of this work are:

- **CAT Framework:** a novel evaluation framework designed to assess the *interplay between accuracy and consistency* of LLM responses, particularly in multiple-choice (MC) settings. This addresses a critical and underexplored non-functional property of LLMs. To support open science, we are releasing the code as open-source at: `https://anonymous.4open.science/r/cora-5B9E/`.

- **CAR Curves, MCA Metric, and CORE Index:** a set of consistency-augmented metrics and tools: the *Consistency-Accuracy Relation (CAR)* curves leverage the *Minimum-Consistency Accuracy (MCA)* metric to quantify accuracy under varying consistency thresholds; and the *Consistency-Oriented Robustness Estimate (CORE)* index, a global metric which summarizes the trade-off between consistency and accuracy.

- **Practical Demonstration with Models and Benchmarks:** a comprehensive evaluation using 8 LLMs across 4 benchmarks, demonstrating how CAT can be used in practice to guide model selection and reveal consistency issues which traditional accuracy metrics tend to overlook.

## 2   RELATED WORK

Consistency of LLM responses (or lack thereof) is becoming a more common research topic in the evaluation of such models, and we can find an increasing number of recent works focused on this topic (Patwardhan et al., 2024; Lee et al., 2024; Habba et al., 2025; Nalbandyan et al., 2025; Ouyang et al., 2025). We also found some notable papers focused on evaluating the consistency of LLM responses, such as (Pezeshkpour & Hruschka, 2024) and (Wang et al., 2024), which explored the impact of either reordering or modifying the set of alternatives for multiple-choice (MC) questions, providing strong evidence that high-performing LLMs may still lack consistency under such variations of prompt that are meaning-equivalent. In (Habba et al., 2025), the DOVE dataset was released with similar semantically-equivalent variations of MC questions but with a more set of input perturbations, such as by varying instruction prompts, separators, in-context samples, among many others, also showing the impact that semantically-equivalent modifications can cause on benchmarks scores. And Pinhanez et al. (2025) showed that even when the exact same input prompt is used, issues with consistency behavior can also be observed given the non-determinism of LLM inference algorithms and tools.

Considering that consistency is a desirable property for LLMs and that there is strong evidence the lack of it is a real issue for many systems, another notable body of research work focuses on providing solutions to either measure or mitigate inconsistent behavior of LLMs. In this group we can cite the SCORE framework (Nalbandyan et al., 2025), which introduces a consistency rate metric based on statistically robust accuracy scores as well as other accuracy-related metric. The work in (Patwardhan et al., 2024) proposes the Majority Voting (MV) metric to define as correct only samples where correct response is pointed out by the majority of alternative MC evaluations. And Raj et al. (2025) presents the Chain of Guidance method to remediate inconsistent behavior of LLMs with enhanced prompts and data augmentation for fine-tuning. Mitigating this problem is beyond the scope of this work.

This work bridges a gap in evaluating response consistency, that is, the interplay between consistency and accuracy of LLMs. As shown in (Pinhanez et al., 2025), both dimensions seem to work as a trade-off, since effects in one dimension often produce the opposite impact in the other dimension.

## 2.1 Notation and Commonly Used Metrics

The standard evaluation method for multiple-choice (MC) benchmarks, referred to as MCQA in this work, is based on single-shot accuracy, i.e. the proportion of correctly answered questions out of a total of $N$ questions, computed after a single pass of LLM inference. Formally, let $Q = \{q_1, \ldots, q_N\}$ be a set of questions, and for each $q_i$, let $a_i$ denote the predicted answer and $o_i^*$ the correct option. Considering the indicator function $\Lambda(.)$, which returns 1 if the condition is correct and 0 otherwise, the MCQA score is then defined as:

$$\text{MCQA} = \frac{1}{N} \sum_{i=1}^{N} \Lambda(a_i = o_i^*) \tag{1}$$

With growing concerns about model consistency, a common strategy is to evaluate LLMs using divergent sets of questions, i.e. variations of the original items which should ideally elicit the same correct answer. We denote this as $Q^* = \{\hat{Q}_1^*, \ldots, \hat{Q}_N^*\}$, where each $\hat{Q}_i^* = \{q_i^1, \ldots, q_i^M\}$ represents a divergence set for sample $i$, potentially including the original question. For each variant $q_i^j$, let $a_i^j$ denote the predicted answer. Formally, let $\text{RC}_i$ denote the response consistency for question $q_i$, defined as:

$$\text{RC}_i = \frac{1}{M} \sum_{j=1}^{M} \Lambda(a_i^j = o_i^*) \tag{2}$$

Prior work has leveraged such divergence sets to define more robust evaluation metrics for LLMs (Pezeshkpour & Hruschka, 2024; Wang et al., 2025). One such metric, MCQA+, computes the average accuracy across all divergent samples:

$$\text{MCQA+} = \frac{1}{N} \sum_{i=1}^{N} \text{RC}_i \tag{3}$$

While MCQA+ incorporates input variation, it provides only a global average, thereby underutilizing the notion of consistency. An alternative is the Majority Voting (MV) metric (Pezeshkpour & Hruschka, 2024), which considers a response correct if the most frequently chosen answer among the $M$ variants matches the ground truth. Let $v_{o,i} = \sum_{j=1}^{M} \Lambda(a_i^j = o)$ be the number of times option $o$ is selected for sample $i$. The MV score is computed as:

$$\text{MV} = \frac{1}{N} \sum_{i=1}^{N} \Lambda\left(\arg\max_{o} v_{o,i} = o_i^*\right) \tag{4}$$

Although MV explicitly accounts for consistency, it applies a relatively permissive threshold and may overlook more nuanced consistency effects, issues explore further in the next section.

## 3 The CAT Framework

Our proposed CAT framework is a set of metrics designed for LLMs based on their behavior concerning the interplay between accuracy and consistency when subjected to input variations. The core of the framework is based on the *Minimum-Consistency Accuracy (MCA)* metric; the *Consistency-Accuracy Relation (CAR)* curves, which serve both to characterize, visualize LLM behavior, and to derive metrics which enable objective evaluation; and the *Consistency-Oriented Robustness Estimate (CORE)* index.

### 3.1 Minimum-Consistency Accuracy (MCA)

A central component of our framework is the tunable *Minimum-Consistency Accuracy (MCA)* metric which evaluates the accuracy of an LLM on a multiple-choice (MC) benchmark. It considers correct only those responses which meet or exceed a specified minimum consistency threshold $c$. Formally,

let $\mathrm{RC}_i$ denote the response consistency for sample $i$, as defined in Equation 2, and let then $c$ be the minimum acceptable consistency level. Then, the MCA at threshold $c$ is defined as:

$$\mathrm{MCA}(c) = \frac{1}{N} \sum_{i=1}^{N} \Lambda(\mathrm{RC}_i \geq c) \tag{4}$$

The metric $\mathrm{MCA}(c)$ is particularly useful when evaluating accuracy at specific consistency levels, which can be, for instance, requirements defined by some regulation for the use of LLMs in high-stakes domain. A notable case is $\mathrm{MCA}(1.0)$, which reflects the proportion of responses that are both correct and fully consistent. However, high values of $c$ may overly penalize the model, while lower values may resemble majority voting (MV), which tends to overestimate accuracy at moderate to high consistency levels, as we show in Figure 5. Consequently, interpreting the values of the metric is depending on the value of $c$ and it can be difficult to characterize a holistic behavior of a model over different values for this parameter.

## 3.2 CONSISTENCY-ACCURACY RELATION (CAR) CURVES

To address the interpretability challenges of MCA, posed by the varying $c$ parameter, we propose the *Consistency-Accuracy Relation (CAR)* curve, which visually represents how accuracy evolves as consistency requirements become increasingly stringent. Such curves represent the evolution of accuracy as consistency requirements vary from the very permissive extreme case with MCA(0.0) to the very penalizing opposite extreme case with MCA(1.0).

A CAR curve is constructed by varying the consistency threshold parameter $c$ over a defined range $[c_{\min}, c_{\max}]$, and computing, for each value of $c$, the corresponding $\mathrm{MCA}(c)$ score. This process yields a curve which captures the inverse relationship between increasing consistency requirements and the resulting accuracy, as measured by $\mathrm{MCA}(c)$ (see figure 1). Formally, let $C = \{c_1, \ldots, c_K\}$ be a set of consistency thresholds, where $c_1 = c_{\min}$, $c_K = c_{\max}$ and $c_k < c_{k+1}$ for all $0 \leq k < K$. The CAR curve is then defined as the set of points:

$$\mathrm{CAR} = \{(c_k, \mathrm{MCA}(c_k)) \mid c_k \in C\} \tag{5}$$

The CAR curve serves mostly as a visualization tool for comparing models based on the shape and area under their respective curves, as in the case of synthetic models shown in figure 1. In the next section, we introduce a CAR-based metric which summarizes this evaluation into a single index.

## 3.3 CONSISTENCY-ORIENTED ROBUSTNESS ESTIMATE (CORE) INDEX

We propose the *CORE* index to evaluate how closely a model's CAR curve aligns with the behavior of a *perfect* model, that is, one which maintains perfect accuracy across all consistency thresholds, represented by a horizontal line at $y = 1.0$. To achieve this, CORE combines two components: the area under the CAR curve and the similarity of the curve to the ideal line.

We start by defining the *Area Under the CAR Curve* (AUCAR) as:

$$\mathrm{AUCAR} = \int_{c_{\min}}^{c_{\max}} \mathrm{MCA}(c) \, dc \tag{6}$$

In practice, this integral is approximated numerically using methods such as the trapezoidal rule over the discretized set $C$:

$$\mathrm{AUCAR} \approx \sum_{k=1}^{K-1} \frac{\mathrm{MCA}(c_k) + \mathrm{MCA}(c_{k+1})}{2} \cdot (c_{k+1} - c_k) \tag{7}$$

While AUCAR provides a scalar summary of the consistency and accuracy trade-off, it does not capture how closely the model's behavior follows the ideal consistency pattern. To address this, a shape-based similarity measure using Dynamic Time Warping (DTW) is incorporated into the index. In details, consider that the ideal CAR curve is a horizontal line at $y = 1.0$, indicating perfect

accuracy at all consistency levels. To quantify the similarity between the model's CAR curve and this ideal curve, we compute the inverse of DTW distance between them, normalized with the maximum possible DTW distance represented by the distance between the ideal curve and the curve of the worst possible model, i.e. when $MCA(0) = 1$ and $MCA(c > 0) = 0$.

Let $DTW_{model}$ denote the DTW distance between the model's CAR curve and the ideal line, and let $DTW_{worst}$ represent the distance between the worst-case curve and the ideal line. We define the normalized DTW similarity as:

$$norm\text{-}DTW = 1 - \frac{DTW_{model}}{DTW_{worst}} \tag{8}$$

Finally, the *CORE* score is computed as the product of AUCAR and the norm-DTW similarity:

$$CORE = AUCAR \times norm\text{-}DTW \tag{9}$$

This formulation ensures that the index $CORE \in [0, 1]$, where higher values indicate both strong overall performance and a CAR curve which closely resembles the ideal line. The CORE index thus serves as a comprehensive indicator of LLM robustness under consistency requirements. Figure 2 shows the values of the CORE index for random-answering synthetic models with positive biases varying from $+0.10$ to $+1.0$. Notice that CORE is very sensitive to the inconsistency of random answering and only reaches high values as the corresponding CAR curves (shown in figure 1) increase both in height and convexity.

## 4 EMPIRICAL VALIDATION OF THE CAT FRAMEWORK

In this section we present an exemplary implementation and validation of our proposed framework using real-world LLMs and multiple-choice (MC) benchmarks. For this, we conducted a study using LLMs in the medical domain, a critical area where inconsistencies in LLMs can pose significant risks. Specifically, we evaluated four *expert* LLMs fine-tuned on medical data alongside their corresponding *base* models, which served as their initialization. Both expert and base models were assessed across four benchmarks: one domain-specific and three general-purpose for validation. This setting allows us not only to evaluate our metrics across a diverse set of models and domains, but also the effects of finetuning.

### 4.1 EXPERIMENT DETAILS

The expert LLMs considered were MedLlama3 7B (MedL) Medical (2024), BioMedical Llama3 8B (BMedL) Medical (2024), BioMistral 7B (BMist) Labrak et al. (2024), and MedAlpaca 7B (MAlpa) Han et al. (2025). Their respective base models are Llama3.1 8B Aaron Grattafiori et al (2024), Llama3.1 8B Instruct Aaron Grattafiori et al (2024), Mistral 7B Instruct Jiang et al. (2023), and Llama 1 7B Touvron et al. (2023).

For domain-specific evaluation, we used MedQA Jin et al. (2020), comprising 1,273 five-choice questions from USMLE exams. For general-domain evaluation, we employed MMLU-Redux 2.0 Gema et al. (2024), ARC-Challenge (ARC) Clark et al. (2018), and TruthfulQA (TruthQA) Lin et al. (2022). MMLU-Redux includes 5,700 questions with 4 choices; ARC contains 1,172 questions with 4 to 5 choices; and TruthQA consists of 817 questions with 2 to 12 choices.

All benchmarks are multiple-choice, allowing for objective evaluation. Each question was formatted using the following prompt, where $QUESTION and $CHOICES were replaced accordingly. The model's response was parsed to extract the selected choice, which is then compared to the correct answer.

```
Answer the following multiple choice questions. The first line of your
response should be: 'LETTER' (without quotes), followed by a step-by-step
explanation.

Question: $QUESTION$
Choices: $CHOICES$
Answer:
```

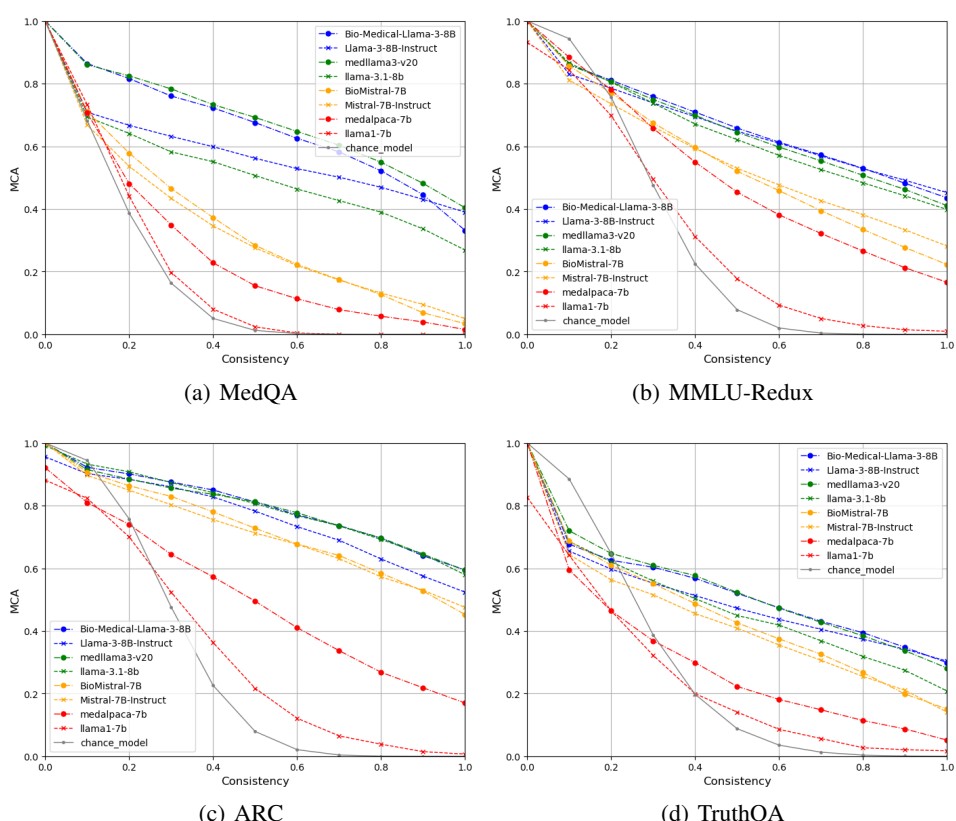

|     |     |
| :-: | :-: |
| (a) MedQA | (b) MMLU-Redux |
| (c) ARC | (d) TruthQA |

Figure 3: CAR curves across benchmarks.

To generate the set of divergent questions $Q^*$, we reordered the answer choices, a well-known and impactful method for inducing inconsistency in LLMs Wang et al. (2025); Pezeshkpour & Hruschka (2024). We created 10 reordered versions per question to compute CAR curves with sufficient granularity. Since we manipulated only the input (not inference parameters like temperature), we use greedy decoding exclusively to avoid inference inconsistency. For ARC, we applied 25-shot in-context learning, a common practice for this dataset Aaron Grattafiori et al (2024); DeepSeek-AI & Aixin Liu et al (2024), while in the other benchmarks we used 0-shot prompting. In-context examples were selected based on similarity to the test sample.

To establish a baseline, we included a *chance* model which randomly selects an answer with uniform probability. This process is repeated 10 times per question for 1,000 samples, and repeated 100 times. The average score of the 100 resamplings was used as the chance baseline which is included as a reference in Figures 1, 2, and 5.

## 4.2 ANALYSIS OF CAR CURVES AND METRIC BEHAVIOR

Figure 3 presents the CAR curves across the four benchmarks, showcasing the performance of all eight LLMs (detailed results are available in Section A). Figure 4 illustrates the progression of four main evaluation metrics, i.e. two from the state-of-the-art (MCQA+[1] and MV) and the two metrics proposed in this work, MCA(1.0) and CORE, along with the AUCAR and norm-DTW (simply DTW in the plots) to show each of these components individually. The former reflects strict consistency (100% agreement) while the latter captures both the area and shape of the CAR curve.

The CAR curves in Figure 3 reveal distinct consistency-accuracy profiles among the LLMs. Low-performing models tend to exhibit concave, steadily decreasing curves, while high-performing ones exhibit convex patterns. Notably, some models, such as Llama-1, perform close to the chance baseline.

---

[1]We omit MCQA due to its high correlation with MCQA+

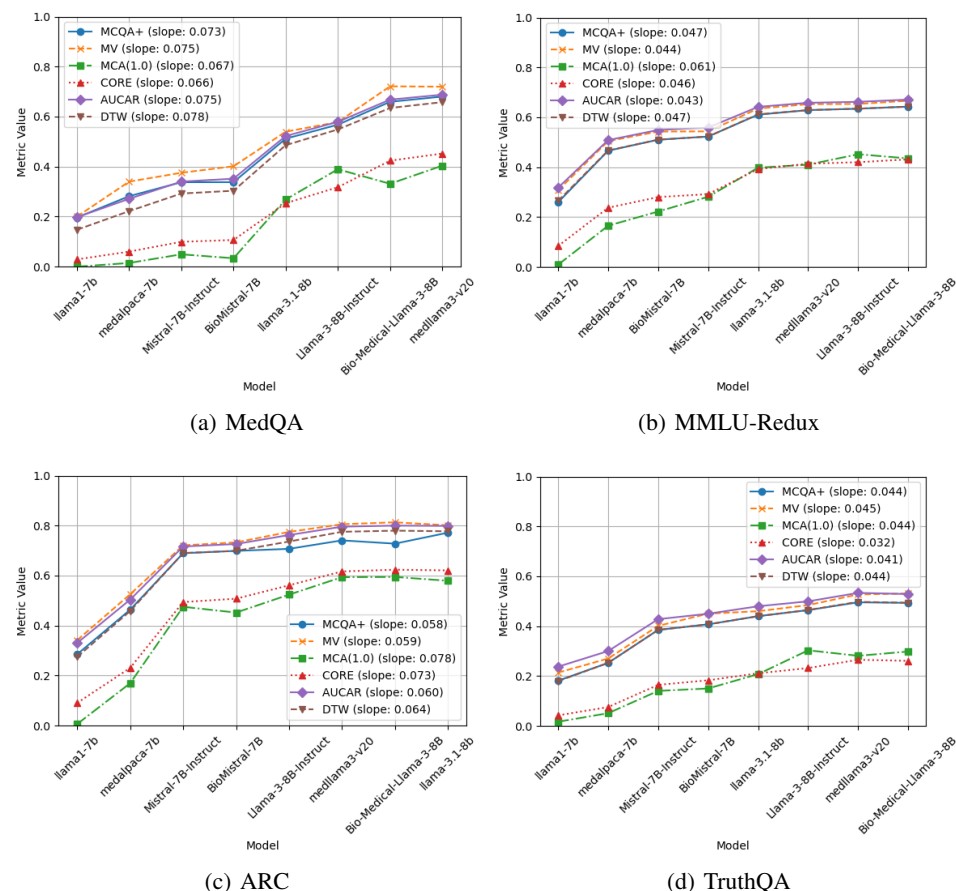

| (a) MedQA | (b) MMLU-Redux |

| (c) ARC | (d) TruthQA |

Figure 4: Metric growth curves, sorted by mean score across all metrics. Legends indicate the slope from chance to best-performing model.

The curves also highlight potential negative biases for consistency levels below 0.2: most models under-perform the chance baseline (except on MedQA) and, in some cases, even at higher consistency levels (e.g., Llama-1 and Medalpaca on TruthQA). In addition, CAR curves can be effective in distinguishing different consistency-accuracy interplay behavior, such as with Llama-3-8B-instruct and Bio-Medical-Llama-3-8B in MedQA; and Llama-3.1-8b and Llama-3-8B-instruct in TruthQA. In both cases, Llama-3-8B-instruct stands out on very strict consistency requirements, i.e. $c = 1.0$.

The CAR curves also expose a beneficial side effect of domain-specific finetuning. First, models that behave considerably different before finetuning, such as Llama-3.1-8b and Llama-3-8B-instruct, tend to present more similar CAR curves after finetuning. Another aspect that becomes evident is the improved performance of expert models on general-domain benchmarks. This is particularly evident for weaker models like Llama-1 and MedAlpaca, where the latter consistently outperforms the former on ARC and MMLU-Redux. This might be attributed to the presence of health-related content in these general benchmarks. While MCQA+ and MV scores reflect this improvement, CAR curves provide a more nuanced view, especially in distinguishing models from the chance baseline.

The metric growth curves in Figure 4 show that CORE and MCA(1.0) generally exhibit more gradual growth when compared to MCQA+ and MV, with slopes of 0.059, 0.048, and 0.031 on MedQA, MMLU-Redux, and TruthQA, respectively. This indicates that consistency-aware metrics are less prone to early saturation, offering better differentiation among models. Moreover, CORE and MCA(1.0) seem to be strongly correlated, except in specific cases such as Llama-3-Instruct on MedQA, where MCA(1.0) suggests a preference over BioMedical-Llama-3 under strict consistency requirements. This kind of correlation has been observed before in the simpler context of multi-repetitions of prompts without variations (Pinhanez et al., 2025). Moreover, it shows that both CORE

and MCA can be viable alternatives to evaluate models considering the consistency aspect together with accuracy, but the choice between one or another is dependent on context and consistency-level requirements. Lastly, notice that AUCAR and DTW, individually, tend to present overly-estimated scores close to MV and MCQA+, while their combination tunes down such scores to being closer to what is presented by MCA(1.0).

We believe these results show the value of the proposed CAT framework by demonstrating that CAR curves and the derived MCA and CORE metrics offer a novel, more refined, and interpretable perspective for the evaluation of the trade-offs between consistency and accuracy of LLMs. These tools not only support model selection based on consistency requirements, but also reveal behavioral patterns such as negative biases and performance proximal to chance. The framework seems to be especially valuable in high-stake domains like healthcare, where reliability is critical and where the use of similar curve-based metrics is common. By capturing the consistency-accuracy trade-off, CAT can complement traditional metrics which we believe often saturate and overlook behavioral stability. Overall, this contribution advances the broader goals of building more robust, transparent, and trustworthy language models.

## 5 EXTENDING CAT TO OPEN-ENDED BENCHMARKS

While our analysis thus far has focused on multiple-choice (MC) benchmarks, the CAT framework is inherently general and can be extended to open-ended evaluation settings. This extension requires a soft adaptation of the $\Psi$ and $\Lambda$ functions introduced in Section 2.1.

In the open-ended case, correctness is no longer binary but instead measured by the similarity between the model-generated response and a reference ground-truth answer. Under this formulation, MC evaluation can be seen as a special case where the $\Psi$ function returns 1 for exact matches (i.e., correct answers) and 0 otherwise. To generalize this, we replace the binary decision with a continuous similarity score, such as cosine similarity (Raj et al., 2025), BLEU score (Cavalin et al., 2025), or even LLM-as-a-judge approaches (Gu et al., 2025), normalized to the $[0, 1]$ range. Similarly, the indicator function $\Lambda$ should also be adapted for the case where response consistency is viewed as the compactness of the cluster of responses instead of the ratio of correct responses, where we can use the same similarity function to compute such compactness ratio.

Provided that the chosen similarity function reliably reflects semantic correctness, all metrics proposed in this work (e.g., CAR curves, MCA, and CORE) can be directly applied to open-ended tasks. This generalization enables the use of CAT in a broader range of evaluation scenarios, including summarization, free-form QA, and generative reasoning tasks.

## 6 CONCLUSION AND FUTURE WORK

In this work, we proposed the CAT framework, a novel approach for evaluating consistency LLMs in relation to accuracy. Through a series of synthetic and benchmark-based experiments, we showed that CAT effectively captures diverse behavioral patterns in LLMs, offering opportunities for a richer understanding of LLMs' performance beyond traditional metrics which consider accuracy and consistency as unrelated dimensions.

For future work, several directions are worth exploring. First, it is important that we validate the framework on a broader and more diverse set of benchmarks and models to enhance the generalizability and robustness of our findings. Second, while we outlined how the framework can be extended to open-ended tasks, implementing and validating this generalization is an important next step. Third, improving the interpretability of the CORE index is essential to make consistency-aware evaluation more accessible and actionable for practitioners. This may involve incorporating normalization strategies, leveraging chance baselines, or designing user-friendly visualizations.

Ultimately, we regard the CAT framework as a step towards more transparent, reliable, and nuanced evaluations of LLMs, particularly in applications where consistency is as critical as correctness.

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

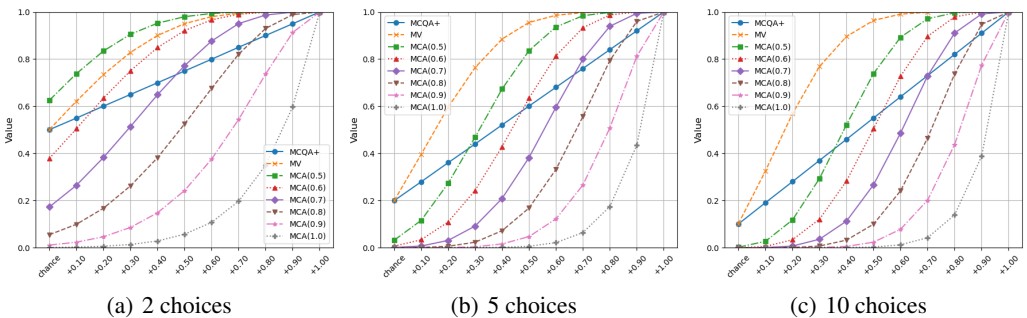

| (a) 2 choices | (b) 5 choices | (c) 10 choices |

Figure 5: Similar to Figure 2 but with additional curves for MCA, showing that this metric tends to approximate MV as the consistency parameter decreases.

## A    DETAILED RESULTS

Figure 5 presents a demonstration of different values of the parameter $c$ in the MCA metric, compared with MCQA+ and MV.

Table 1 depicts the complete results for the LLM evaluations described in Section 4.2.

| Dataset | Model | MCQA | MCQA+ | MV | MCA(1.0) | CORE |
|---|---|---|---|---|---|---|
| MedQA | Bio-Medical-Llama-3-8B | 0.703 | 0.659 | **0.721** | 0.332 | 0.424 |
| | Llama-3-8B-Instruct | 0.566 | 0.568 | 0.578 | 0.390 | 0.318 |
| | medllama3-v20 | **0.735** | **0.680** | 0.720 | **0.404** | **0.452** |
| | llama-3.1-8b | 0.504 | 0.512 | 0.540 | 0.269 | 0.254 |
| | BioMistral-7B | 0.336 | 0.338 | 0.401 | 0.034 | 0.107 |
| | Mistral-7B-Instruct | 0.306 | 0.338 | 0.375 | 0.049 | 0.100 |
| | medalpaca-7b | 0.261 | 0.282 | 0.340 | 0.015 | 0.060 |
| | llama1-7b | 0.189 | 0.197 | 0.200 | 0.000 | 0.029 |
| | *chance | 0.200 | 0.200 | 0.200 | 0.000 | 0.023 |
| MMLU-Redux | Bio-Medical-Llama-3-8B | **0.660** | **0.643** | **0.667** | 0.435 | **0.432** |
| | Llama-3-8B-Instruct | 0.651 | 0.635 | 0.654 | **0.452** | 0.420 |
| | medllama3-v20 | 0.644 | 0.629 | 0.653 | 0.410 | 0.414 |
| | llama-3.1-8b | 0.623 | 0.611 | 0.635 | 0.398 | 0.393 |
| | BioMistral-7B | 0.520 | 0.510 | 0.543 | 0.222 | 0.280 |
| | Mistral-7B-Instruct | 0.539 | 0.523 | 0.544 | 0.282 | 0.292 |
| | medalpaca-7b | 0.486 | 0.466 | 0.505 | 0.166 | 0.238 |
| | llama1-7b | 0.262 | 0.260 | 0.308 | 0.009 | 0.084 |
| | *chance | 0.250 | 0.250 | 0.250 | 0.000 | 0.075 |
| ARC | Bio-Medical-Llama-3-8B | 0.743 | 0.728 | **0.813** | **0.595** | **0.624** |
| | Llama-3-8B-Instruct | 0.714 | 0.707 | 0.775 | 0.524 | 0.561 |
| | medllama3-v20 | 0.738 | **0.741** | 0.805 | 0.594 | 0.617 |
| | llama-3.1-8b | **0.774** | 0.772 | 0.801 | 0.579 | 0.621 |
| | BioMistral-7B | 0.705 | 0.699 | 0.733 | 0.452 | 0.508 |
| | Mistral-7B-Instruct | 0.692 | 0.690 | 0.721 | 0.475 | 0.494 |
| | medalpaca-7b | 0.456 | 0.464 | 0.526 | 0.170 | 0.231 |
| | llama1-7b | 0.304 | 0.286 | 0.342 | 0.007 | 0.091 |
| | *chance | 0.250 | 0.250 | 0.250 | 0.000 | 0.075 |
| TruthQA | Bio-Medical-Llama-3-8B | **0.542** | 0.494 | **0.531** | 0.299 | 0.261 |
| | Llama-3-8B-Instruct | 0.536 | 0.465 | 0.485 | **0.304** | 0.232 |
| | medllama3-v20 | 0.541 | **0.496** | 0.528 | 0.282 | **0.266** |
| | llama-3.1-8b | 0.414 | 0.440 | 0.460 | 0.208 | 0.211 |
| | BioMistral-7B | 0.318 | 0.408 | 0.450 | 0.151 | 0.183 |
| | Mistral-7B-Instruct | 0.244 | 0.385 | 0.401 | 0.141 | 0.165 |
| | medalpaca-7b | 0.266 | 0.253 | 0.271 | 0.051 | 0.076 |
| | llama1-7b | 0.065 | 0.182 | 0.214 | 0.017 | 0.043 |
| | *chance | 0.226 | 0.226 | 0.226 | 0.000 | 0.062 |

Table 1: Evaluation metrics for each model grouped by dataset. Best values per metric per dataset (higher is better) are in bold.

