# OpenReview forum: "CAT: A Metric-Driven Framework for Analyzing the Consistency-Accuracy Relation of LLMs under Controlled Input Variations"
_ICLR.cc/2026/Conference — Submitted to ICLR 2026_

### Official Review · Reviewer_Dg53 · 2025-10-26

**Soundness:** 2
**Presentation:** 3
**Contribution:** 2
**Rating:** 2
**Confidence:** 4

**Summary:**

The authors proposed the CAT framework for evaluating the consistency–accuracy relationship of large language models (LLMs). This framework introduces an indicator function that checks whether the proportion of correct answers exceeds a specified threshold and extends the current majority-vote baseline. The final CORE metric is defined as the product of (a) the integrated performance across different threshold levels and (b) the model’s performance scaled between the best and worst models.

**Strengths:**

1. The proposed framework is straightforward and easy to understand, while offering a more comprehensive evaluation compared to the traditional majority-vote approach.
2. The authors conducted extensive experiments across eight models and four datasets, demonstrating the robustness and validity of the proposed metric.

**Weaknesses:**

1. The authors should provide a clearer justification for taking the product of two CAR-derived metrics that both range from 0 to 1 (Eq. 9), as this operation implicitly introduces correlation and amplifies the scale. From Figure 4, the DTW and AUCAR metrics appear very similar; including some extreme cases where either metric fails would help clarify their distinct contributions.
2. The simulation of accuracy–consistency results is based on a single prompt, which limits the strength of the conclusions. Incorporating multiple prompt rephrasings would better assess the robustness and generalizability of the proposed metric.
3. While evaluating thresholds from 0.5 to 1.0 is reasonable since the model is accurate in this range, the paper would benefit from a stronger motivation for analyzing consistency in the 0 to 0.5 range—specifically, why evaluating consistency for lower-accuracy cases is meaningful.

**Questions:**

1. It is unclear whether the CORE metric has an interpretable scale. For instance, whether the worst performance consistently corresponds to a specific baseline value (e.g., X), and how an increase of the CORE metric to Y translates into a measurable performance improvement (e.g., Z%). Clarifying this interpretability would strengthen the practical utility of the metric.

---

### Official Review · Reviewer_LChs · 2025-10-30

**Soundness:** 2
**Presentation:** 2
**Contribution:** 2
**Rating:** 2
**Confidence:** 4

**Summary:**

The paper presents CAT, a method to visualize the relation between accuracy and response consistency via Consistency Accuracy Relation (CAR) curves. The paper also shows experiments with CAT over MC benchmarks with general and domain-specific models.

**Strengths:**

1. The CAR curves seem to capture an interesting dimension regarding stochastic LLM evaluation.

**Weaknesses:**

1. The writing is not clear enough.
2. There are redundant parts, e.g., eq. 6 and 7, the evaluation prompt.
3. Overall, the contribution is suited for a full paper.
4. There is a lack of a bigger story behind the results. No clear problem or gap was stated that the paper solves.
5. A bit of odd notation, use an indicator function. The CAR curve is continuous, yet it is presented as discrete.
6. The experimental setup is unclear. Why consider llama-1?
7. It is not clear what information regular metrics do not provide that CAT adds. How can it change model selection?.
8. The extension to open-ended questions can benefit from experiments that realize it.

**Questions:**

Try to understand what the bigger picture is here. What someone that do not use your metric misses. Understanding it and framing the paper accordingly, together with further investigating this direction, can significantly improve the paper.

---

### Official Review · Reviewer_sZ9F · 2025-11-02

**Soundness:** 3
**Presentation:** 3
**Contribution:** 3
**Rating:** 4
**Confidence:** 3

**Summary:**

This paper introduces CAT (Consistency–Accuracy Toolkit), a metric-based framework for jointly evaluating accuracy and response consistency of large language models (LLMs) under controlled input perturbations, using multiple-choice (MC) benchmarks as a case study.
The framework defines:
- MCA (Minimum-Consistency Accuracy), measuring accuracy conditioned on a minimum consistency threshold c;
- CAR (Consistency–Accuracy Relation) curves, plotting accuracy against consistency requirements;
- CORE (Consistency-Oriented Robustness Estimate), combining the CAR curve’s area and shape similarity (via normalized DTW) into a single score.
Experiments on eight LLMs and four benchmarks (including MedQA, MMLU-Redux, ARC, and TruthfulQA) show how the framework can reveal consistency–accuracy trade-offs that conventional metrics (e.g., MCQA+, Majority Voting) overlook.

**Strengths:**

- The paper tackles an important and underexplored dimension of LLM evaluation, i.e. the trade-off between accuracy and consistency, which is critical for model trustworthiness and real-world deployment.
- The definitions of MCA, CAR, and CORE are mathematically well specified and easy to reproduce. The CAR curves provide an intuitive visualization analogous to calibration or precision–recall plots.
- The study includes diverse models (general-purpose and medical-domain) and benchmarks, illustrating that the framework can be applied across contexts.

**Weaknesses:**

1. While the framework is well-executed, its methodological novelty is limited. MCA essentially generalizes the majority voting (MV) metric with a tunable threshold; CAR curves are simply MCA(c) plotted over varying thresholds, which is a standard evaluation pattern; CORE combines the area (AUC) and a DTW-based shape similarity, which feels like an engineering aggregation rather than a fundamentally new evaluation principle.
2. The use of DTW to measure curve similarity appears unnecessarily complex and mathematically redundant in this setting.
Since the ideal curve (y=1) and worst curve (step-like 1->0) are both nearly flat, the DTW distance reduces to a point-wise vertical deviation integral, effectively equivalent to measuring the area under the curve (AUC):

$DTW_{model,ideal} \approx \sum_i |1 - MCA(c_i)| \Rightarrow \text{norm-DTW} \approx AUC_{CAR}$

offering little new information beyond the AUC itself.

The paper should clarify whether DTW yields any non-trivial discrimination compared to simpler measures such as mean absolute deviation or correlation with the ideal curve.

3. The paper defines the worst-case curve as MCA(0)=1 and MCA(c>0)=0, which is conceptually problematic (correct me if I'm wrong):
- By definition, MCA(0)=1 holds for all models since every sample trivially satisfies RC>=0; this point carries no discriminative information.
- Including it artificially reduces the normalization denominator DTW_{worst}, thereby slightly inflating all normalized scores.

A more principled and interpretable definition would set CAR_{worst}(c)=0 for all c, corresponding to total failure across all consistency thresholds.
This would simplify normalization (DTW_{worst}=1) and make norm-DTW easier to interpret as a direct distance from the ideal line.

4. While the experiments cover multiple datasets and models, the analysis remains qualitative:
- Only choice-order shuffling is used as perturbation;
- No analysis of semantic, instructional, or contextual variations.

**Questions:**

1. What empirical advantage does DTW offer compared to simpler vertical deviation or correlation-based measures?
2. Would redefining the “worst model” curve as y=0 materially change CORE rankings?
3. Have you validated that CORE rankings correlate with human judgments of reliability or stability?

---

### Official Review · Reviewer_qqLB · 2025-11-03

**Soundness:** 2
**Presentation:** 2
**Contribution:** 3
**Rating:** 4
**Confidence:** 2

**Summary:**

This paper proposes CAT (Consistency–Accuracy Toolkit), a framework for evaluating large language models (LLMs) by jointly analyzing accuracy and response consistency under controlled input perturbations. Using multiple-choice (MC) benchmarks as a testbed, the authors define three main components: (i) Minimum-Consistency Accuracy (MCA), which measures accuracy conditional on achieving at least a minimum consistency threshold $c$; (ii) Consistency–Accuracy Relation (CAR) curves, which visualize how accuracy changes as consistency requirements tighten; and (iii) CORE, a global robustness index that multiplies the area under the CAR curve (AUCAR) by a DTW-based shape-similarity term comparing each model’s curve to an ideal flat line at 1.0.

The authors apply CAT to eight models—including both generalist and medical-domain finetuned LLMs—across four benchmarks (MedQA, MMLU-Redux, ARC, and TruthfulQA). They find that stronger models exhibit more convex CAR curves, reflecting higher stability under stricter consistency thresholds, and that domain-specific finetuning improves consistency on in-domain tasks. The paper argues that CAT provides a more nuanced picture of model reliability than standard single-pass accuracy metrics like MCQA+ or Majority Voting (MV) and that the framework can, in principle, be extended to open-ended tasks through similarity-based scoring.

**Strengths:**

(1) Timely focus on consistency and robustness. \
The paper addresses an important and underexplored dimension of LLM evaluation: the stability of model predictions under semantically equivalent perturbations. This focus is especially timely given growing concerns about data leakage, contamination, and benchmark overfitting. Controlled perturbation and consistency analysis can help reveal whether a model has genuinely learned the underlying task or simply memorized patterns from training data. A systematic framework for measuring such effects is therefore valuable for the broader evaluation community.

(2) Clear formalization of metrics. \
The formal definitions of MCA(c), CAR, and CORE are mathematically precise and easy to reproduce. The CAR formulation elegantly unifies consistency and accuracy by conditioning correctness on different levels of consistency, providing a rigorous basis for further extensions. The framework could plausibly be applied beyond multiple-choice settings by substituting alternative similarity functions, making it a flexible foundation for future work.

(3) Visualization as a conceptual tool. \
The CAR curve offers a clear visual abstraction for examining the relationship between model accuracy and consistency. While not perfectly intuitive on first read, the concept has potential to help practitioners reason about stability under perturbation once better interpretive guidance is provided.

**Weaknesses:**

(1) Clarity and organization of exposition (Figures 1–2; Section 3). \
The paper is difficult to follow on an initial read, largely because key figures and metrics are introduced out of order. Figures 1 and 2, showing synthetic CAR curves and the proposed CORE metric, appear in the introduction before any of the underlying metrics (MCQA+, MV, MCA, CORE) are defined. This sequencing makes the early figures challenging to interpret without the accompanying metric definitions, which could be introduced earlier for clarity. The introduction would benefit from a high-level explanation of what each metric captures (e.g., "MCA controls for a minimum consistency level $c$, CORE summarizes the trade-off as area × shape similarity") before any formal equations are presented. While I was ultimately able to follow the definitions and interpret the results, I found the presentation unusually dense, and it’s possible that my difficulty stems in part from not being deeply familiar with prior work on consistency-aware LLM evaluation metrics. Even so, I believe the organizational and explanatory issues would challenge most readers who are not already steeped in this area.

(2) Limited intuition and motivation for MCA(c) and CORE. \
The paper would benefit from clearer intuition about why the proposed metrics—particularly MCA(c) and CORE—are needed beyond existing baselines like MCQA+ and Majority Voting (MV). From my understanding, the main difference between MCQA+ and MCA(c) is that MCQA+ averages the proportion of correct answers across all perturbations, whereas MCA(c) measures the proportion of questions whose per-item accuracy exceeds a minimum consistency threshold $c$. In other words, MCA(c) converts a continuous measure into a thresholded one, which changes the aggregation but not obviously the underlying concept being measured.
Figures 1–4 help illustrate some general properties of the metrics, but the paper would be stronger if it explicitly walked through these relationships. For instance, it would help to note directly that MV and MCA(0.5) are closely related: MV checks whether the majority of perturbed responses are correct, while MCA(0.5) asks whether at least 50% of the variants yield the correct answer—a slightly more formalized version of the same principle. The small divergence between the two for low-bias models (as seen in Figure 2) likely arises because MV can still count a case as correct when the correct answer is the modal prediction even if it appears fewer than half the time—especially when incorrect responses are diverse—whereas MCA(0.5) requires a strict majority of correct responses. Making this intuition explicit would help readers understand why MCA(c) is more conservative for weak or inconsistent models and how it generalizes existing metrics rather than merely reweighting accuracy across perturbations.
A similar clarification would help for CORE. In Figure 2, MCQA+ increases linearly with model bias, while CORE rises more slowly for near-chance models and steepens for higher-bias ones. It seems plausible that this non-linear shape reflects the multiplicative structure of CORE—combining the area-under-curve (AUCAR) and a DTW-based shape-similarity term—but it is not entirely clear from the text.

(3) Limited intuition for the norm-DTW component of CORE. \
The paper could provide more intuition for the role of the norm-DTW term in the definition of CORE. From Figure 1, it appears that the “ideal” CAR curve corresponds to a horizontal line at 1.0 (the perfectly consistent, perfectly accurate model). Even simply pointing this out in the text could help readers interpret what the shape comparison is meant to represent. The norm-DTW term is described as measuring the distance between this ideal curve and a model’s actual CAR curve, thereby penalizing deviations in shape. However, it is not immediately clear what distinct information this shape penalty contributes beyond what is already captured by AUCAR, which already summarizes the area under the curve. Intuitively, one would expect smaller AUCAR values to correlate with greater deviation from the ideal flat line, since a model with low area necessarily lies farther below 1.0. If norm-DTW is intended to capture something else—such as curvature or irregularity in how consistency degrades with $c$—it would be helpful to articulate that explicitly.

(4) Unclear correspondence between synthetic and empirical CAR-curve behavior. \
The relationship between the synthetic CAR curves in Figure 1 and the empirical curves in Figure 3 could be explained more clearly. In Figure 1, the synthetic models biased toward the correct answer (e.g., bias = 0.9) exhibit strongly convex CAR curves, showing a rapid rise in accuracy at low $c$ that quickly plateaus near 1.0. By contrast, in Figure 3, the best-performing real LLMs show only mild convexity and in many cases appear nearly linear. It is therefore not obvious why the shapes of the empirical curves differ from those of the synthetic ones, even though both are supposed to represent models with high effective bias toward the correct answer. Clarifying why real LLMs produce smoother or more linear CAR curves—and what that implies about their underlying consistency dynamics—would help connect the synthetic illustration to the empirical findings.

(5) Ambiguity in interpreting under-chance performance at low consistency thresholds. \
The paper notes that several models “under-perform the chance baseline for consistency levels below 0.2.” Could the authors clarify what this means? My understanding is that at very low thresholds, a random (chance) model can appear to perform better because it occasionally answers some variants correctly by luck, whereas an LLM that consistently selects the same incorrect answer across all perturbations would receive a lower MCA(c) score.

(6) Ambiguity in the cause of improved strict-consistency performance. \
The paper notes that models such as Llama-3-8B-Instruct outperform their base or domain-specific counterparts at very strict consistency levels (e.g., $c = 1.0$) on MedQA and TruthfulQA. However, it is unclear what drives this effect. The text attributes it to domain-specific finetuning, but another plausible explanation is that these models have simply seen more multiple-choice–style data during instruction tuning—becoming more familiar with the format and with answer-choice permutations—rather than benefiting from exposure to the specific domain content. Both hypotheses could explain improved consistency under stringent thresholds, but the current experiments do not distinguish between them.

(7) Unclear interpretation of expert-model gains on general-domain benchmarks. \
The paper observes that domain-specific expert models (e.g., MedAlpaca) outperform their base counterparts (e.g., Llama-1) on general-domain benchmarks such as ARC and MMLU-Redux, suggesting that domain finetuning may sometimes improve broader performance. However, without further evidence, this interpretation feels speculative. It is not obvious why medical-domain finetuning should transfer to more general reasoning tasks, and the observed effect may instead reflect overlap in benchmark content or idiosyncrasies in dataset composition—particularly given that MMLU-Redux includes some health-related categories. The authors briefly acknowledge that this may be due to such dataset characteristics, but given that caveat, this observation should not be framed as a substantive empirical result in the main analysis. Clarifying this distinction would make the discussion of the findings more accurate and appropriately scoped.

(8) Need for further analysis to disentangle domain alignment from format familiarity. \
The stronger strict-consistency performance (e.g., $c = 1.0$) of models like Llama-3-8B-Instruct is an interesting finding, but it remains unclear whether this improvement primarily reflects domain exposure or simply greater familiarity with the multiple-choice format—such as increased exposure to question–answer pairs and answer-order variations during finetuning. Both factors could plausibly improve consistency under stringent thresholds, but the current setup does not distinguish between them. An ablation that separates domain-specific finetuning from multiple-choice format exposure—for instance, by finetuning one model on domain-matched but non-MC data and another on general but MC-heavy data—would clarify which factor most strongly contributes to high-consistency behavior.

(9) Limited perturbation diversity; results may be permutation-set dependent. \
The current evaluation focuses on answer-order permutations. While this is a meaningful first step, it risks conflating the framework’s behavior with the particular permutation family used. In practice, consistency can fail for many other meaning-preserving perturbations (e.g., paraphrasing the stem, altering instruction style or tone, toggling rationales, changing separators or formatting, injecting benign distractors, varying few-shot exemplars). Different perturbation families can induce different error structures (e.g., correlated vs. uncorrelated mistakes), which will in turn change CAR shapes and derived metrics (MCA(c), AUCAR, CORE). Evaluating at least one or two additional perturbation types beyond answer order—or reporting sensitivity of results to the number and type of variants $M$—would clarify whether the framework’s conclusions reflect general consistency properties of the models or artifacts of a single permutation design.

**Questions:**

1. Could the authors explicitly explain why MCA(0.5) and Majority Voting differ for low-bias models but converge for higher-bias ones? Clarifying this connection in the text would make the intuition behind MCA(c) much clearer.

2. Beyond area differences, what qualitative behavior does norm-DTW capture that AUCAR does not? Are there specific cases where two models have similar AUCAR but different DTW distances, and what behavioral difference would that reflect? More generally, why was Dynamic Time Warping chosen as the measure of shape similarity? Were other notions of curve-shape similarity considered, and what motivated DTW as the preferred choice?

3. Why do empirical CAR curves for strong LLMs appear more linear or only mildly convex compared to the sharply convex synthetic high-bias curves in Figure 1? Could this reflect correlated error structures or some other aspect of the consistency behavior demonstrated by real LLMs? I found this difference interesting and think it warrants more explanation or investigation.

4. The paper notes that several models "under-perform the chance baseline for consistency levels below 0.2". Could the authors clarify what this means in concrete terms—perhaps by including qualitative examples? My understanding is that at low thresholds, a random model may appear stronger because it occasionally answers some variants correctly by luck, whereas an LLM that consistently selects the same distractor across all perturbations would receive a lower MCA(c) score. It would be helpful to clarify whether this observation arises mainly in low-$K$ benchmarks like MedQA and ARC, where random guessing has a higher probability of producing such "lucky" partial consistency. More generally, is this truly a negative property of the LLMs? If a model is consistently wrong in a systematic way—e.g., always choosing the same distractor—then under-chance behavior might not indicate instability but rather a consistent bias.

5. The higher MCA(1.0) for instruction- or domain-finetuned models is an intriguing result, but its cause is ambiguous. Is this improvement primarily driven by domain exposure, or could it instead stem from greater familiarity with the multiple-choice format itself—such as increased exposure to question–answer pairs and answer-order variations during finetuning? Either hypothesis could explain why these models perform better under strict consistency thresholds. To determine which explanation holds, an ablation separating domain-specific finetuning from MC-format familiarity would be particularly worthwhile. For instance, finetuning one model on domain-matched but non-MC data and another on general but MC-heavy data could clarify which factor most strongly drives high-consistency behavior.

6. The current evaluation focuses exclusively on answer-order permutations. Evaluating at least one or two additional perturbation types beyond answer order—or reporting sensitivity of results to the number and type of variants $M$—would clarify whether the framework’s conclusions reflect general consistency properties of the models or artifacts of a single permutation design. This limitation makes the current results less convincing; additional experiments here would meaningfully increase the paper’s impact and persuasiveness.

---

### Meta-Review · Area_Chair_7THA · 2025-12-03

**Summary:**

This paper proposes to study the interplay between accuracy and response consistency.  Reviewers raised the following criticisms: (1) poor presentation, (2) limited motivation for proposed metrics, (3) lack of fine-grained analysis of any interesting findings, for example the stronger strict-consistency performance of models like Llama-3-8B-Instruct, which may require ablations, (4) focus on only answer-order perturbation, (5) limited novelty, (6) unnecessary and poorly motivated complexity.  All of these criticisms are significant.  There were many other minor criticisms raised by reviewers which I am omitting here.

**Reviewer Concerns:**

As there were no rebuttals, all feedback was not addressed.

**Reviewer Scores:**

Reviews started out at 4, 4, 2, 2.  As there were no rebuttals, I assume all reviewers would maintain their scores.  The reviewers unanimously vote for rejection, and I agree with the reviewers regarding most of their points.  Therefore, I am also recommending rejection at this time.

---

### Decision · Program_Chairs · 2026-01-26

Reject